# Quantifying Information without Entropy: Identifying Intermittent Disturbances in Dynamical Systems

**DOI:** 10.3390/e22111199

**Published:** 2020-10-23

**Authors:** Angela Montoya, Ed Habtour, Fernando Moreu

**Affiliations:** 1Sandia National Laboratories, Albuquerque, NM 87185, USA; acmont@sandia.gov; 2William E. Boeing Department of Aeronautics & Astronautics, University of Washington, Seattle, WA 98195, USA; habtour@uw.edu; 3Department of Civil, Construction, & Environmental Engineering, University of New Mexico, Albuquerque, NM 87131, USA

**Keywords:** information entropy, discontinuity detection, intermittent disturbance, nonlinear dynamical system

## Abstract

A system’s response to disturbances in an internal or external driving signal can be characterized as performing an implicit computation, where the dynamics of the system are a manifestation of its new state holding some memory about those disturbances. Identifying small disturbances in the response signal requires detailed information about the dynamics of the inputs, which can be challenging. This paper presents a new method called the Information Impulse Function (IIF) for detecting and time-localizing small disturbances in system response data. The novelty of IIF is its ability to measure relative information content without using Boltzmann’s equation by modeling signal transmission as a series of dissipative steps. Since a detailed expression of the informational structure in the signal is achieved with IIF, it is ideal for detecting disturbances in the response signal, i.e., the system dynamics. Those findings are based on numerical studies of the topological structure of the dynamics of a nonlinear system due to perturbated driving signals. The IIF is compared to both the Permutation entropy and Shannon entropy to demonstrate its entropy-like relationship with system state and its degree of sensitivity to perturbations in a driving signal.

## 1. Introduction

The detection of small transient events, momentary internal or external discontinuities in the inputs or outputs, in nonlinear dynamical systems has received considerable attention in recent years. It continues to be an open scientific problem in many disciplines such as physics [1], engineering [2], and biological sciences [3]. Dynamical systems can be thought of as networks of interacting elements that change over time [4,5,6,7,8,9]. Those systems display complex changing behaviors at the global (macroscopic) scale, emerging from the collective actions of the interacting elements at the microscopic scale [10,11]. Due to the complexity of nonlinear systems, small global or microscopic transient events can influence the emerging dynamics and trajectories of those systems [2]. A natural propensity for chaotic behavior can exist in any complex system or network [12]. Thus, it is crucial to develop efficient methods for detecting small transient events and gain insights into the ramification of those events by tracking the evolution of the system dynamics. Examples of transient events may include malicious attacks on social or communication networks [13], whirled in sand particles instigated by tiny disturbances [14], intermittent switching between open- and closed-cracks in mechanical structures leading to sudden cascaded failure [15], acoustic flame extinction [16], or evidence of a distant catastrophic event like an earthquake [17].

Detecting discontinuities can be particularly challenging in systems exhibiting nonlinear, chaotic, or time-varying behaviors [18,19,20,21]. Although effective in many cases, detection methods that are based on signal statistics ignore causal or correlational relationships between data points. For example, the mean and standard deviation are typically calculated with the assumption that each data point is independent and part of the same population. If the assumption is reasonable for most data points, i.e., any relationship between points is weak, then outliers can be detected by merely noting points that are significantly different from the mean. However, transient events that correlate with the dynamics of the system produce subtle changes to the shape of the distribution at the tails. This subtlety can mask an unusual behavior if the data are not partitioned or not modeled appropriately [22].

For a dynamical system, consecutive points are limited to physically realizable solutions of the underlying governing equations of the system. In other words, consecutive data points in a dynamical system are inherently causal, so sudden disturbances will change the system trajectory if only for a short amount of time [23]. As a result, the system will have a topological structure intrinsic to the dynamic behaviors instigated by the disturbances. The topological structure holds some memory of the disturbances, which are reflected in the dynamics of the new state of the system [23,24]. Memory, however brief, implies the storage or transference of information [7]. In recent years, measures of information have become an important tool for discriminating between deterministic and stochastic behavior in dynamical systems [25,26].

The concept that physical systems produce information was proposed to the dynamics community by Robert Shaw [27]. His idea revolutionized the approach to dynamical systems analysis by allowing the use of information theory, namely entropy, to quantify changes in a dynamical system’s topological structure. Information entropy measures the variation of pattern in a sequence, thereby capturing the internal structure or relationships that exist between data points [28]. However, information entropy, like its thermodynamic counterpart, describes macroscopic not microscopic behaviors. Just as with signal statistics, small or localized system changes can be easily engulfed by larger or more dominant behaviors.

This paper aims to solve the technical challenge of creating a reliable method for detecting small discontinuities in the response signal of a nonlinear dynamical system even during chaos. This is particularly important since small disturbances in chaotic systems may have large consequences. To do this, the authors chose to take an information theorical approach to account for changes in topological structure of the system. Some background on dynamical systems and information theory is provided to the reader in Section 2 to explain this approach. To overcome the limitation of using statistically-based methods to examine changes in response data that are statistically insignificant, an entropy-like function without the need of using Boltzmann’s equation was devised. Discussion and derivation of the new function are provided in Section 3. The function appears to reveal a detailed informational structure of the response signal that, as demonstrated by the example in Section 4, is sensitive to small intermittent events or impulses. Therefore, the authors call it the Information Impulse Function (IIF), which examines a response signal for discontinuity events on a more localized scale than traditional information entropy methods. Numerical experiments were then conducted to examine the detection sensitivity of the IIF. Results in Section 5 show that the IIF outperforms Permutation entropy and Shannon entropy in detecting small discontinuities in a signal and is able to capture the change in the dynamic structure of the system using the Poincare distance plot, which is the standard for characterizing the nonlinear and chaotic behaviors in dynamical systems [3]. Finally, the conclusion and future work are presented in Section 6.

## 2. Theory and Background

### 2.1. Topological Structures of Dynamical Systems

In this section, a general overview of the theory and applicability of nonlinear dynamical system is provided, with an emphasis on the states behaviors of a Duffing system [29]. The Duffing system was chosen because of its rich complexity, considerable utility in physics and engineering applications, and as a benchmark in analyzing nonlinear systems. From a mathematical perspective, the topological structure of any dynamical system can be referred to as the system state space, S, such that η(τ)∈S defines the current dynamic states of the interacting elements in the system. Evolving from a present to a future state in a continuous time system is detected by an evolution law that acts on S. Typically, the data or signal collected from a dynamical system, and the law acting on S can be expressed as a system of differential equations (Equation (1)):(1)η ˙= F(η,τ), η∈S, τ∈ℝ

A state space S can be a set of discrete elements, a vector space, or a manifold. The vector η(τ) represents the state of the dynamical system at a nondimensional characteristic time τ, where η is having dimension n≫1. The function 𝓕(·) is the evolution law that represents the dynamics of the data. To improve the sensitivity of detecting momentary disturbances in a dynamical system, it is necessary to describe its topological changes due to small discontinuities in the input signal (global disturbances), and the parameters (internal disturbances) of the system. To this end, the Duffing system, which consists of a cubic nonlinear term, can be represented as a nonlinear ordinary differential equation (ODE), as follows:(2)$$\ddot{\eta }\;=\;-\;2\zeta \;\dot{\eta }\;+\;\eta \;-\;k\eta^{3}\;+\;E\left(\Omega ,\tau \right)$$

The forcing function E(z,τ) can be thought of as an input signal into a dynamical system. A harmonic signal, where E(Ω,τ)=Fcos(Ωτ), is selected in this study. The forcing amplitude and frequency are F and Ω, respectively. The dissipative and nonlinear restoring forces contain a damping ratio, ζ, and a cubic coefficient, k, respectively. The system’s natural frequency, ω, is equal to one, such that the linear restoring force is ωη = η. The over-dots represent the temporal derivatives for the system response η, which can be output signals at time τ. In this paper, the dynamical system in Equation (2) is treated as a mechanical system. Therefore, $\dot{\eta }$ and $\ddot{\eta }$ are referred to as velocity and acceleration, respectively. The coefficients in Equation (2) are nominally set to the dimensionless values: ζ = 0.15, k = 1, Ω = 1.2. These values were chosen as they produce a wide range of dynamical behavior over a relatively short range in excitation values. The range used in this paper was 0.1 to 0.45.

The topological evolution of a dynamical system can be visualized via Poincaré distance [30], which is based on the system phase portrait, i.e., velocity verses response, as shown in Figure 1. The Poincaré distance reveals important intrinsic features about the dynamics of the system such as its stability, nonlinearity, and chaotic behaviors. The Poincaré distance, r, is computed with Equation (3), were η and $\dot{\eta }$ are taken at intervals of Δτ=0.45. The parameter r is the Euclidian distance from the origin on a phase diagram for η(τ):(3)$$r\;=\;\sqrt{{\left(\eta \right)}^{2}\;+\;{\left(\dot{\eta }\right)}^{2}}$$

From the Poincaré plot, it can be observed that the system displays steady state behavior as long as 0<F<0.29. As the excitation amplitude increases beyond F= 0.29, the system transitions to a chaotic regime and remains there for 0.29<F<0.35. At this point, the system will return to a state of periodic behavior then to a chaotic state for F<0.37.

It is important to point out that the cubic nonlinear term and time-dependent force are responsible for the bifurcation and chaotic behaviors [29]. Under periodic excitation, the system is known to possess multiple steady-state solutions. For example, varying the forcing amplitude or frequency can cause a steady-state dynamic behavior to changes drastically due to a transition from one stable condition to another stable or unstable one. Bifurcation occurs in a system when a small perturbation in a parameter causes a sudden change in the dynamics of the system, as shown in Figure 1 near F=0.26, or 0.29. Transiting from a stable to unstable state can occur due to perturbations (global change), or due to a drift in parameters (local bifurcations). Bifurcations can also lead to chaotic responses, as shown in Figure 1 when F=0.30 to 0.34.

### 2.2. Information Theoretic Descriptions of Topological Structures for Dynamical Systems

Claude Shannon, the founder of information theory, utilized the concept of entropy from statistical thermodynamics to develop a generalized framework for communications architecture [28]. Despite being a single scaler value, Shannon’s Entropy can be utilized to quantify the internal structure or regularity in a signal, which makes entropy an important tool for studying the complexity of and hidden features in a signal.

In statistical thermodynamics, the distribution of energy levels of gas particles in a fixed volume can be estimated using the Boltzmann distribution, which maximizes thermodynamic entropy [31]. Analogous to estimating the frequency of gas particle interactions by maximizing entropy, information theory applies the Boltzmann distribution to model redundancy in a finite sequence of symbols [32]. Given a sequence M of length N with independently selected messages $m_{j}$ that each have a corresponding probability given by $p_{j}$, the normalized Shannon entropy (SE) for M can be calculated as follows:(4)$$H\left(M\right)=\;-\frac{1}{\log_{10}\left(N\right)}\overset{N}{\underset{j=1}{\sum }}p_{j}\log_{10}p_{j},\;for\;j=1,\ldots ,N$$

The redundancy in a sequence M is calculated by categorizing each $m_{j}$ into one of N possible categories, which is referred to as *states* henceforth. By inspection of Equation (4), it can be inferred that SE results are highly dependent on both the number of states and the criteria for assigning of $m_{j}$ to a particular state as that determines the probabilities $p_{j}$. In classical information theory, states are defined by an encoding scheme used to package a signal and transmit it across a channel [28]. The main criterion for avoiding information loss is that the encoding scheme must contain enough dimensionality to uniquely describe all possible states. Implicit in Shannon’s derivation is that states have significance to the analyst. The examples in his seminal work use words and letters in English sentences to make that point. The analog for response data of a mechanical or biological system, however, is ambiguous. Significance is an arbitrary condition, which means the definition of a state and the set of all possible states is open to the interpretation of the analyst.

As a consequence, there are many extensions of SE, in which they retain the utility of Boltzmann’s distribution: $H=\;-\sum^{\;}pLog\left(p\right)$ and address the concept of state for a dynamical system in different ways. Notable examples relevant to this paper are Kolmogorov–Sinai entropy, Permutation entropy, spectral entropy, and Singular Entropy. Unlike the extensions mentioned above, the IIF, as will be explained in the next section, produces comparable results even though the use of Boltzmann equation is explicitly abandoned.

Kolmogorov–Sinai entropy, also known as metric entropy, defines states as partitions in phase space, making a direct connection to the physics of dynamical systems [33,34]. A reconstruction of the phase space is required for the computation of the entropy, which can make metric entropy impractical for real-time applications [35].

Permutation entropy was introduced to account for the complexity of order in a sequence [36]. Unlike metric entropy, permutation entropy does not require a sequence to be the result of a physical process. The states are defined from a set of all possible permutations for a given segment length. Despite permutation entropy disconnection from the physics of the system, the general symbolic equality of permutation entropy and metric entropy was demonstrated, which implies that permutation entropy can be employed to study physical systems [37]. Since permutation entropy is simpler computationally than metric entropy it is widely utilized in many applications. Garland et al., for example, used a rolling permutation entropy to examine anomalies in paleoclimate data [38]. Yao et al. cited the computational ease of permutation entropy as an attractive representative entropy method for quantifying complex processes [39].

Spectral entropy was devised for periodic, stationary signals that the Fourier spectrum was applied to partition the trajectory of a signal into repeatable behaviors [40]. Spectral entropy is commonly used in a variety of applications for signal classification. Recent applications include structural health monitoring [41], and identification of sleep disorders [42]. The continuous utilization of spectral entropy is because a dynamic behavior can be partitioned easily based on its frequency content. The current implementation of the IIF uses the short-time Fourier spectrum (STFT) on this premise (See Appendix A).

Of the four SE extensions, the IIF is most similar to wavelet singular entropy (WSE), which uses the singular values from a singular value decomposition (SVD) of a wavelet time-frequency decomposition in the Boltzmann equation to express an entropy [43]. WSE operates on the premise that the singular values correlate to dynamical states. The IIF assumes that dynamical behavior is expressed in the singular vectors instead. 

For simplicity, the SE probabilities are calculated by binning acceleration amplitude values ($a_{j})$ according to Equation (5):(5)$$p_{j}\;=\;\frac{a_{j}}{\sum_{j}^{N}a_{j}}for\;j=1\ldots N$$

In this study, the total number of bins, N, was 250, which was chosen as a nominally representative and easily computable example of an SE solution given a signal length of 5000 points.

The permutation entropy of order 3, lag 1 (PEn3) is computed by collecting consecutive sequences of three values from a discretely sampled signal, rank-ordering them, and comparing them to a list of six possible patterns: $\pi_{k}\;∋\left\{{\left[1\;2\;3\right]}_{k=1},\;{\left[2\;3\;1\right]}_{k=2},\;\ldots ,\;{\left[2\;1\;3\right]}_{k=6}\right\}$. The signal is incremented by one sample (lag 1), so the patterns overlap. PEn3 is thus computed via Equation (6):(6)$$PEn_{3}\;=\;-\frac{1}{\log_{2}3!}\overset{3!}{\underset{k\;=1}{\sum }}p_{k}\log_{2}p_{k}$$
where $p_{k}$ is calculated by taking the total the number of times a specific pattern, $\pi_{k},$ is observed in the signal and dividing it by the total number of categorized sequences. Details on computing PEn3 can be found in [44]. The permutation entropy results reported in this paper were limited to order 3. Higher orders increased the computation time beyond what would be a reasonable comparison to the IIF without significant gain in sensitivity.

This paper compares the normalized permutation entropy of order 3, lag 1 (PEn3) and an SE solution to the IIF. SE and PEn3 are chosen over Spectral and Singular Entropies because they do not share as many common features with the IIF. Additionally, with the parameters chosen, both of these entropy variations are comparable to the IIF in terms of algorithmic simplicity.

## 3. The Information Impulse Function

Prior to computing the IIF, the signal must be decomposed such that it is uniquely and yet completely described. This is necessary for the SVD computation in the IIF and analogous to encoding the signal for transmission. Vector space representations are common for both dynamical systems analysis and encoding algorithms [45,46,47]. The decomposition method required by the IIF is otherwise not strictly prescribed. Any one of many common time-frequency decomposition methods can represent a time history into an appropriate vector space.

For the purposes of demonstration, a signal f(τ) is decomposed into a vector space using the discrete STFT modulated with a periodic Hann window, W. Equation (7) describes the STFT for f(τ):(7)$$\Upsilon \left(\kappa ,\omega \right)\;=\;\overset{\infty }{\underset{-\infty }{\sum }}f\left(\tau \right)W\left(\tau \;-\;\kappa \right)e^{-i\omega \tau }$$

The decomposed signal f(τ) is represented by Υ, which is a function of two dimensions: frequency ω and discrete time step κ. The resulting size of Υ is m×n, where n corresponds to the number of discrete time steps and m corresponds to the number of frequency bins. In general, this paper assumes n≥m.

Performing an SVD on Υ forms a rotation from the original signal basis to the most compact description of the signal that can be given only by linear transformations. Parenthetically, the SVD description of Υ can be used as an expression of the algorithmic or Kolmogorov complexity of the signal. Equation (8) is the decomposition for Υ, which contains complex values:(8)$$\Upsilon \left(\kappa ,\omega \right)\;=\;USV^{T}$$

The conjugate transpose is V. The singular value matrix, S, is rank ordered such that $S_{1,1}$ > $S_{2,2}$> … > $S_{r,r}$. Υ is a rank C matrix that is equal to the number of nonzero singular values in S, U and V are the left and right singular vector matrixes, respectively. The columns of U and V form a set of orthonormal vectors such that U and V are unitary. Therefore, if Υ is m×n, then U is m× m, S is m×n, and V is n×n, for n≥m.
U and V can be thought of as representation of different behaviors in Υ, which is the signal under study in the frequency domain.

Since $USV^{T}$ is linearly decomposable and rank ordered, Υ can be approximated by rank reduction or SVD truncation [48]. The matrix Υ is approximated by either subtracting out singular values in reverse order starting with the smallest value, or leaving them out of the sum as expressed in Equation (9):(9)$$\Upsilon^{r}\;=\;\overset{r}{\underset{q}{\sum }}US_{q}V^{T}\;q=\;1,\ldots ,r\;r\leq C$$

The resulting rank of the approximation, given by r, is equal to the highest index of the singular value used in the sum. At full rank ( r= C), Υ is approximated by the SVD with minimal error [48]. The net effect on Υ of subsequent approximations by rank reduction is a steady increase in information loss.

The product of the truncated matrix V and the transpose of its complex conjugate $\hat{V}$, and likewise U and $\hat{U}$, only reach the identity matrix I when all vectors (index *j*) are included. Equation (10) shows this by writing the unitary product as a sum similarly to Equation (9):(10)$$\underset{r\to n}{\lim}\overset{r}{\underset{q\;=\;1}{\sum }}V_{iq}{\hat{V}}_{qj}\;=\;I\;\begin{array}{c} i\;=\;1,\ldots ,n\\ j\;=\;1,\ldots ,n \end{array}\;\underset{r\to m}{\lim}\overset{r}{\underset{q\;=\;1}{\sum }}U_{iq}{\hat{U}}_{qj}=\;I\;\begin{array}{c} i\;=\;1,\ldots ,m\\ j\;=\;1,\ldots ,m \end{array}$$

The repeated index does not imply summation, as the sum is shown explicitly.

The intent of IIF is to quantify the *work* performed by the rank reduction on the signal subspaces U and V, and to examine the signal’s behavior. The rate of convergence to the identity matrix for each index i as a function of r is used to represent the information loss by a rank reduction specific to each subspace. This rate can be quantified by collecting the diagonal elements resulting from the sums in Equation (10) for each value of r. The resulting expression is simplified to be the cumulative sum of the conjugate square of the singular vector elements over the index j, which represents different values of r. Subsequently, IIF *potential functions* for U and V are computed in Equation (11):(11)$$\Phi_{ij}^{R}\;=\;\overset{j}{\underset{q\;=\;1}{\sum }}{\left|V_{iq}\right|}^{2}\begin{array}{c} i\;=\;1,\ldots ,n\\ j\;=\;1,\ldots ,n \end{array}\;\Phi_{ij}^{L}\;=\;\overset{j}{\underset{q\;=\;1}{\sum }}{\left|U_{iq}\right|}^{2}\begin{array}{c} i\;=\;1,\ldots ,m\\ j\;=\;1,\ldots ,m \end{array}$$

For each index i, $\Phi_{ij}^{R}$ and $\Phi_{ij}^{L}$ monotonically increase with j. As such, the potentials describe the *contribution* gradient to the full signal with respect to column index j. Alternatively, rank reduction can be thought of as a force acting in the j direction, i.e., performing work. Constraining the direction of action to a single dimension implies that the curl is zero. Without an explicit proof, the expressions in Equation (11) are assumed to represent the conservative potential functions in j for every value of i.

The range for the expressions in Equation (11) (n and m, respectively) are determined by the size of Υ. Unity is reached, as shown in Equation (10), when all vectors are included. However, Equation (11) reaches a maximum in i when j corresponds to a full rank approximation of Υ. This is a consequence of rank ordering in the SVD. The corresponding singular vectors in U and V, must contain redundant information because they are effectively removed from the product. Thus, full-rank approximation of Υ is equated with the maximum information potential. The authors use C to denote this value because the maximum potential can also be thought of as the minimum channel capacity needed to transmit Υ with minimum error. The work can be calculated by integrating the potential functions in the direction *j* up to the maximum potential at C, as follows:(12)$$IIF^{R}\left(i\right)\;=\;\frac{1\;}{C\left(C+1\right)}\overset{C}{\underset{j=1}{\sum }}\Phi_{ij}^{R}\begin{array}{c} i\;=\;1,\ldots ,n\\ j\;=\;1,\ldots ,C\;C\leq m\; \end{array}\;IIF^{L}\left(i\right)=\;\frac{1\;}{C\left(C+1\right)}\overset{C}{\underset{j=1}{\sum }}\Phi_{ij}^{L}\begin{array}{c} i\;=\;1,\ldots ,m\\ j\;=\;1,\ldots ,C\;C\;\leq \;m\; \end{array}$$

Equation (12) implies that the $IIF^{R}$ and $IIF^{L}$ are the work required per index i to compress the signal Υ (compressing gas particles in a thermodynamic system) with respect to the right and left singular vectors, respectively. The $IIF^{R}$ and $IIF^{L}$ are normalized such that the sum over index i for either function is $\frac{1}{2}$. The fact that only a scalar value is needed to normalize Equation (12) is a consequence of defining the potential functions on normalized subspaces. The $IIF^{R}$ and $IIF^{L}$ are given the generic unit of *intensity* since Equation (12) is proportional to square of a vector subspace.

Either the $IIF^{R}$ or the $IIF^{L}$ may be used to interrogate the local information content of a signal. To use the IIF to find a discontinuity in time, this paper uses the $IIF^{R}$ form of the IIF exclusively.

## 4. Application of the IIF to Simulated Dynamic Response Data

### 4.1. Simulation of Discontinuities in Nonlinear Dynamical Systems

The ability of the IIF algorithm to detect discontinuities in system response was examined in this paper by introducing both global and local discontinuities to the system defined in Equation (2). The global discontinuities were injected into the input signal, E(Ω,τ). The local discontinuities, on the other hand, were introduced into the dissipative and nonlinear restoring forces, i.e., damping and stiffness parameters, respectively. Discontinuities were simulated by adding a delta function to the parameter of interest in Equation (2) as follows:(13)$$\left(F_{d},k_{d},\zeta_{d}\right)\;=\;A_{d}\left(F,\;k,\;\zeta \right)\;\left(\delta \left(\tau \;-\;\tau_{i}\right)\right)$$

The impulse amplitude $A_{d}$ is a percentage of the tested parameter F, k, or ζ that varies from 1% to 10% in increments of 1%. the Dirac delta function, δ, is centered at $\tau_{i}$. The result in Equation (13) is zero everywhere until τ reaches $\tau_{i}$. The results presented in this paper are for discontinuities at $\tau_{i}\;=\;160$. Equations (14) through (16) represent the test cases for discontinuities in *force excitation*, *stiffness*, and *damping*:(14)$$\ddot{\eta }\;+\;2\zeta \;\dot{\eta }\;-\;\eta \;+\;k\eta^{3}\;=\;\left(F\;+\;F_{d}\right)cos\left(\Omega \tau \right)$$
(15)$$\ddot{\eta }\;+\;2\zeta \;\dot{\eta }\;-\;\eta \;+\;\left(k\;+\;k_{d}\right)\eta^{3}\;=\;Fcos\left(\Omega \tau \right)$$
(16)$$\ddot{\eta }\;+\;2\left(\zeta \;+\;\zeta_{d}\right)\dot{\eta }\;-\;\eta \;+\;k\eta^{3}=\;Fcos\left(\Omega \tau \right)$$

The above equations were solved numerically using MATLAB ODE45, which is a Runge–Kutta ODEs ordinary differential equation solver. The algorithm was devised to ensure that the outputs can be evenly sampled, and the parameters may be defined as functions in time: ζ(τ), k(τ), F(τ), and Ω(τ). Figure 2 describes the procedure.

In addition to the three test cases described by Equations (14)–(16), a set baseline test cases were made that contained no discontinuities of any kind. Baseline cases were not needed to detect discontinuities with the IIF but were created to provide additional insight into the numeric results presented in the next section.

### 4.2. Example Output

While the IIF can be computed for any input or output variable, $\ddot{\eta }$ was chosen since it was more sensitive to changes than $\dot{\eta }$ or η. Therefore, the results reported in this paper were computed from the output signal $\ddot{\eta }$, which is a common measurement in physics and engineering applications. An example of calculating the IIF from an acceleration signal, $\ddot{\eta }$, due to an input signal, E, with a disturbance, $F_{d}$, while the system was experiencing chaotic behaviors is provided in Figure 3.

The disturbance was simulated as a Dirac delta function with an amplitude $F_{d}\;=\;0.0179$ (10% of F), which was a discontinuity superimposed on the harmonic input signal with an amplitude F= 0.447 at time $\tau_{i}\;=\;160.0$. For comparison, the simulation was also computed without the disturbance and superimposed the case with a disturbance, as shown in Figure 3. It can be observed from Figure 3a that the input signal with disturbance shown in red is exceeding the undisturbed signal in black line is barely visible near τ=160. However, the aftermath of that minute event is evident in the acceleration signal well after the occurrence of the disturbance due to the chaotic nature of the system (Figure 3b). The $IIF^{R}$ with and without the disturbance were calculated using Equation (12) and plotted in Figure 3c. Clearly, the IIF identified the occurrence of and amplified the presence of the disturbance, which was the reason for selecting $F_{d}\;=\;0.1F$ to illustrate the mechanics of IIF. Choosing $F_{d}\;<\;0.1F$ would make it more challenging to show the effect of the disturbance in the acceleration signal since the system was in a chaotic regime. Even though $\ddot{\eta }$ is more sensitive to disturbances than $\dot{\eta }$ or η, the chaotic response of the system can easily camouflage the presence of small discontinuities in the signal.

The application off the IIF in Figure 3c clearly shows the occurrence of the disturbance when its amplitude was 0.10F even when the system was chaotic. The $IIF^{R}$ peak value was 89.9% greater than the mean value of $IIF^{R}$, which was approximately $4.0\times 10^{-4}$. IIF mean value is ½ over the total number of singular vectors in V (Equation (9)). Incidentally, the IIF mean value is a convenient datum as it does not depend on the signal complexity.

However, the total number of vectors in V is a consequence of the parameters used in the STFT decomposition procedure (Equation (7)). One of those parameters is the size of the Hann window in the STFT, which determines the time resolution of the IIF, and influences its sensitivity. Applying a 16-point window the simulation shown in Figure 3. Subsequently, the resulting time resolution in the IIF was 0.2τ, which was coarser than that of the $\ddot{\eta }\;$signal at 0.05τ. Additionally, the STFT caused the energy to distribute or ‘leak’ across multiple time bands as an artifact of sampling, which is one of the well-known shortcomings of decomposition procedures [49]. This choice of initial decomposition method impacts the sensitivity of the IIF and adds uncertainty to the location of the disturbance. The STFT was chosen as a representative decomposition method for this paper because it is one of the most well understood methods and is able to produce IIF results with a good signal to noise ratio. The tradeoff between sensitivity and uncertainty for different decomposition methods is currently under study.

To gain insight into this limitation of the IIF, a closer examination of the results in Figure 3 based on the net effects (baseline-test case) are provided in Figure 4. The data points are marked to highlight the difference in temporal resolution. The net changes due to a disturbance in the input and response signals are shown in Figure 4a,b, respectively. Figure 4c shows the IIF for STFT with 8, 16, and 32 window sizes. The 16-point window appears to be the most effective coverage for a single point discontinuity. Therefore, the results from the experimental simulations provided in this study were likewise based on a Hann window size of 16 points in the STFT prior to computing the IIF.

## 5. Results

The numerical results reported in this section demonstrate the sensitivity of the IIF in detecting momentary global and local discontinuities occurring in a dynamical system with cubic nonlinearity described in Section 2.1. The algorithm was examined for multiple steady-state excitation amplitudes, which were varied from 0.10 to 0.45 by an increment of 0.001. The objective was to examine IIF response to a change in the state of the system regardless of the amplitude level of the input signal. Discontinuities in the (i) input excitation; (ii) stiffness; and (iii) damping were investigated for ten impulse amplitudes in each of the three cases by Equation (13).

The results for the first case are provided in Section 5.1 with a detailed discussion. The results and analyses of the IIF’s ability to detect internal disturbances in the stiffness and damping components, i.e., cases (ii) and (iii), respectively, are reported in Section 5.2.

### 5.1. Detecting Global Disturbances

This section presents IIF computational results for input signals with different forcing and discontinuity amplitudes for the nonlinear dynamical system given in Equation (14). This test case represents situations where the system experiences an external (or *global*) disturbance. For each numerical experiment, the peak $IIF^{R}$ (occurring at τ=160.0 in all trials) was computed and compared to Shannon entropy (SE) and permutation entropy (PEn3) calculations, which were calculated using Equations (5) and (6), respectively.

Figure 5 and Figure 6 show SE and PEn3 results (respectively) for all the experiments performed in case (i) as a function of excitation and discontinuity amplitudes, F and $F_{d}$. At very close examination, SE results showed an ability to differentiate between discontinuity levels by a very small drop in value for some excitation levels. Close examination of $PEn_{3}$ results, however, did not show similar success. For example, the percent change from 1 to 10%
$A_{d}$ for SE was −0.04% at F =0.25 with a steady decrease in values. The percent change from 1 to 10%
$A_{d}$ in peak $IIF^{R}$ peak values for F= 0.25 was 14.9%. PEn3 showed no change.

Figure 7 shows peak $IIF^{R}$ values for the same data set. It can be observed that the IIF peak values monotonically, although not linearly, increased when the discontinuity amplitudes were increased. The $IIF^{R}$ was able to detect the presence of a single point discontinuity clearly in all experiments, i.e., $F_{d}=F\times \left\{0.01,0.02,\;\ldots ,\;0.10\right\}$ including the chaotic region.

It should be clear from Figure 5, Figure 6 and Figure 7 that the IIF is much more sensitive to small discontinuities in the excitation force than either of the more traditional entropy approaches. Neither technique was able to clearly differentiate between 1% or 10% discontinuities in the input signal when the system was chaotic.

Commonalities between the SE, PEn3, and IIF calculations are shown in Figure 8, Figure 9 and Figure 10, respectively. These calculations are superimposed on a 10% discontinuity Poincaré plot to illustrate the detection of the three methods in various stable and chaotic regions.

The three approaches share similar characteristic behavior. In general, they appeared to trend, to some degree, with the state of the system expressed in Poincare’ plots (Figure 8 through Figure 10). Each had clear inflection points near F = 0.15 and 0.26, which was consistent with the Poincaré transitions. Changes in the slopes in all three approaches occurred near or after sharp declines were in trends with the transitions in the Poincaré diagram. Additionally, all three approaches appeared to identify the chaos- stable transition between F≈0.35 to 0.38. This was a strong evidence that the IIF, despite not being derived from Boltzmann’s formula behaves like an entropy with regard to system state.

### 5.2. Detecting Internal Disturbances

This section provides detailed discussion for the (ii) stiffness and (iii) damping cases based on the numerical results for local discontinuities for the nonlinear dynamical systems presented in Equations (15) and (16), respectively. As with the previous section, for each numerical experiment the peak IIF (occurring at τ= 160.0 in all trials) was computed and compared to Shannon entropy (SE) and permutation entropy (PEn3) calculations.

Figure 11 and Figure 12 show SE and PEn3 results (respectively) for all the experiments performed in case (ii) as a function of excitation and stiffness discontinuity amplitudes, F and $K_{d}$. Likewise, Figure 13 and Figure 14 show SE and PEn3 results for all experiments performed in case (iii) as a function of the excitation and stiffness discontinuity amplitudes, F and $\zeta_{d}$. These results are similar to those found in Section 5.1 in both the general shape of the function with respect to excitation level and insensitivity to small discontinuities.

In contrast, IIF peak values in Figure 15 and Figure 16 show the IIF was able to detect the presence of a single point discontinuity clearly in all experiments, i.e., $K_{d}=K\times \left\{0.01,0.02,\;\ldots ,0.10\right\}$ and $\zeta_{d}=\zeta \times \left\{0.01,0.02,\;\ldots ,0.10\right\}$ including the chaotic region. Similar to case (i), the IIF peak values for cases ii and iii monotonically increased when the discontinuity amplitudes were increased.

Another important observation is shapes of the curves in Figure 15 and Figure 16 are significantly different from those in global disturbance case (Figure 7). This is due to the fact that in the IIF is sensitive to the system’s instantaneous response to a discontinuity. In contrast, the SE and PEn3 results for case (ii) shown in Figure 11 and Figure 12, respectively, closely resemble those in case (i). Likewise, SE and PEn3 results for case (iii) (Figure 13 and Figure 14) are also similar to case (i) results.

To more explicitly show the effectiveness of the IIF over SE and PEn3, the percent difference from the largest discontinuity ($0.10A_{d}$ from Equation (13)) to the smallest discontinuity ($0.01A_{d}$) was calculated by Equation (17):(17)$$\%\;Difference=100\frac{0.10\;A_{d}\;-0.01\;A_{d}\;}{0.10\;A_{d}\;}$$

In Figure 17, the % difference for results from cases (i), (ii) and (iii) are shown for IIF peak values in (a), SE in (b), and PEn3 in (c). The effectiveness of the IIF over SE and PEn3 can be concluded from the large difference in scale on the percent difference coordinate between Figure 17a to those of Figure 17b,c, respectively. Additionally, both SE and PEn3 approaches have percent differences with both positive and negative signs. This suggests that the SE and PEn3 approaches are not only insensitive to small discontinuities they are unreliable as a detection method when the system is chaotic.

## 6. Discussion

A new method to analyze small, momentary internal or external disturbances in time history data has been presented called the Information Impulse Function (IIF). The IIF is derived by solving for the notional work done to transmit a signal that has been decomposed or represented in a vector space. The IIF methodology provides a way to estimate localized information content in a signal without the a priori information needed to rely on the Boltzmann distribution explicitly. 

Numerical experiments were conducted to test the ability of the IIF to detect single-point discontinuities in a nonlinear system. Disturbances between 1% and 10% of the nominal parameter amplitudes were introduced into the excitation, stiffness, and damping terms. The IIF was able to detect all momentary internal and external disturbances in the response signal even when the system was chaotic. The results suggest that the amplitude of the IIF was sensitive to both the background complexity of the system and the change in system response. Topological changes in the dynamic behavior of the system due to small discontinuities can be identified easily by tracking the IIF peak values. This was true for when the source of the disturbance was external or internal. The IIF behavior followed similar trends as both Shannon and permutation entropy, suggesting that the IIF behaved like an entropy. However, the IIF was far more superior in detecting small disturbances than permutation entropy and Shannon entropy.

The ability of IIF to detect and localize small discontinues in an input to a dynamical system can be useful feature in many biological, medical, scientific, and engineering applications. Investigating the robustness of the IIF to random noise as well as studies on the utility of the IIF for source identification are on-going research activities. Future work also includes research into the use of alternate decomposition methods.

## Figures and Tables

**Figure 1 entropy-22-01199-f001:**
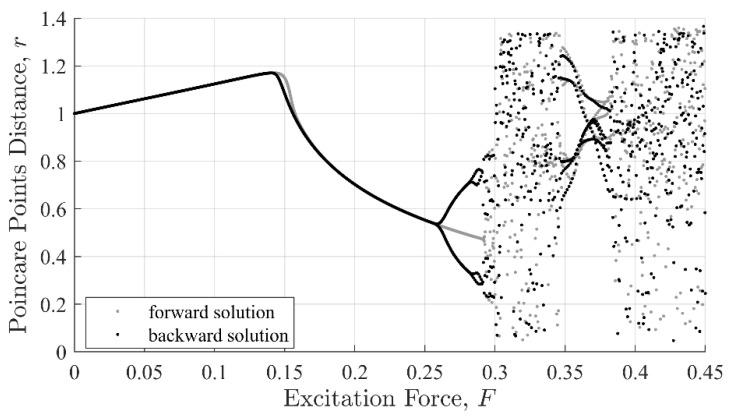
Poincaré distance showing the topological evolution of the Duffing oscillator with parameters: ζ=.15, k=1, and Ω=1.2. Forward and backward solutions are given in grey and black, respectively.

**Figure 2 entropy-22-01199-f002:**
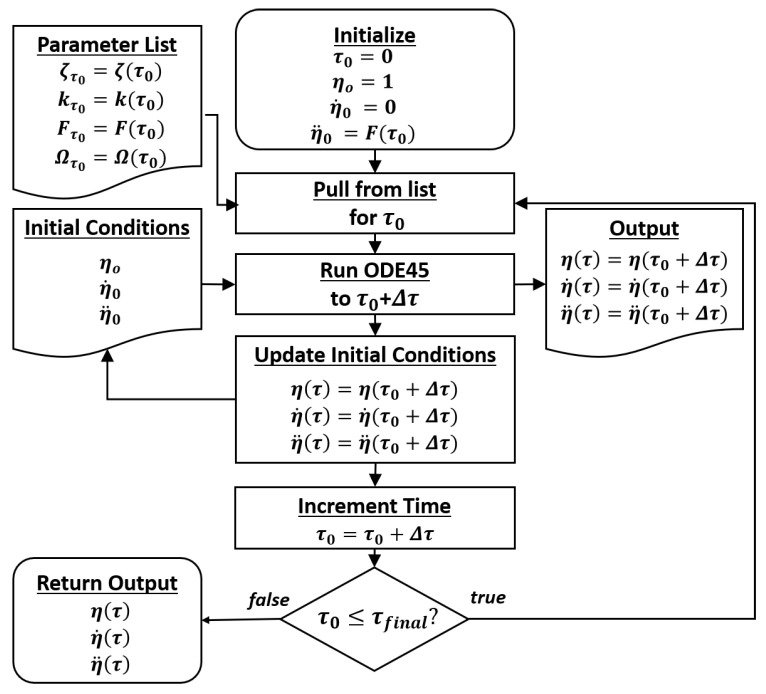
Algorithm used to simulate a Duffing oscillator for variable parameters.

**Figure 3 entropy-22-01199-f003:**
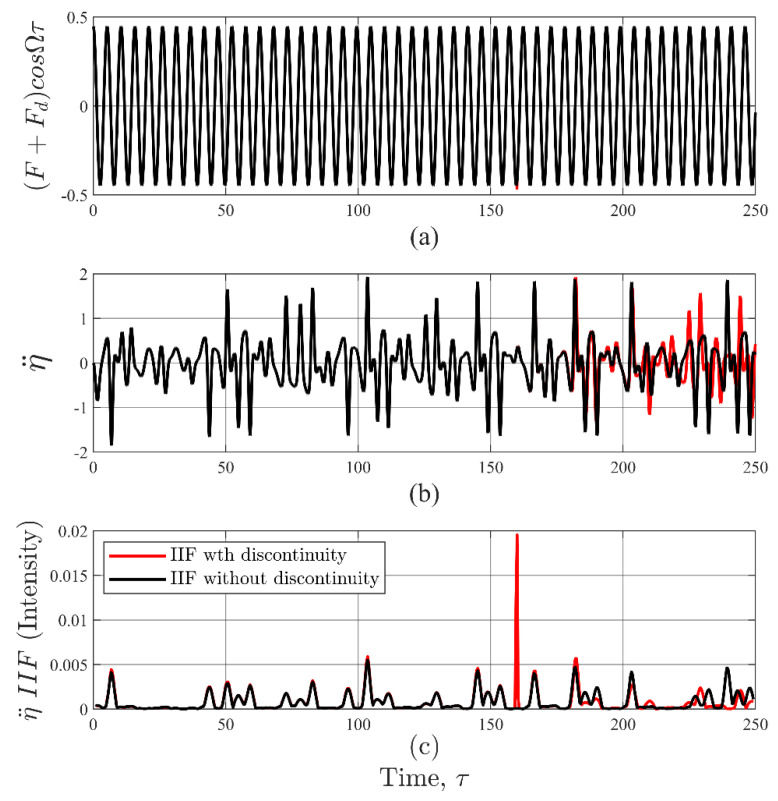
Example IIF analysis on the acceleration time history of Duffing oscillator in chaos. Plotted in black are results without a discontinuity. Plotted in red are results containing a discontinuity. (**a**) The sinusoidal excitation. The discontinuity at τ = 160 is almost not visible. (**b**) The acceleration response. Due to the chaotic nature of the system, the effect of a discontinuity is evident for τ > 160 when compared to the response without a discontinuity. (**c**) The IIF results. The IIF computed on response data containing a discontinuity produced a peak value near τ = 160 that easily distinguishable from the background. In contrast, the IIF computed on the response data without a discontinuity did not produce any large peaks.

**Figure 4 entropy-22-01199-f004:**
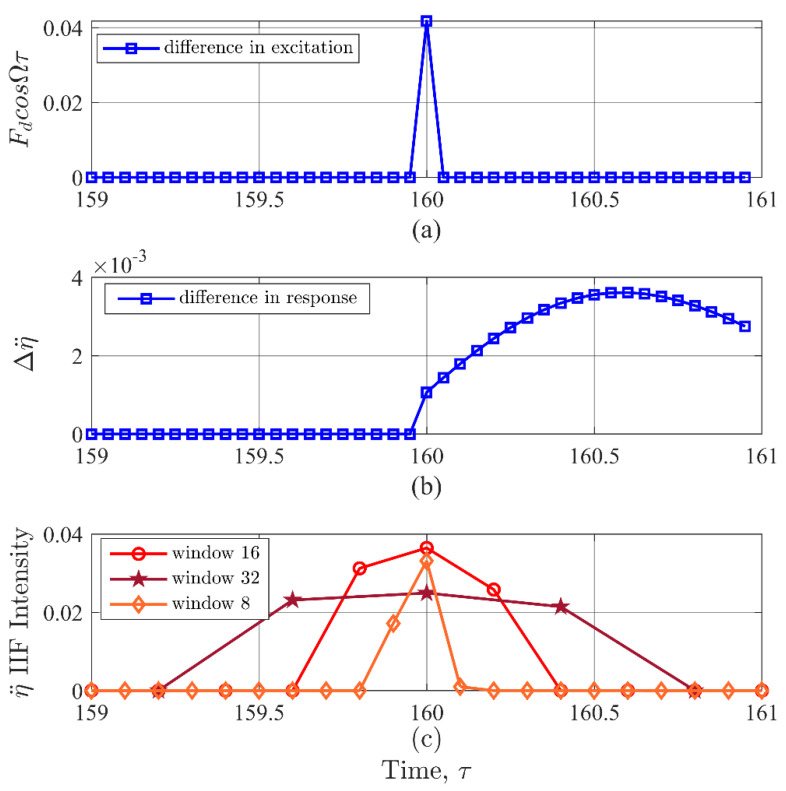
(**a**) Difference in excitation from the baseline to the test case. This figure shows the precise location of the discontinuity. (**b**) Difference in response from the baseline to the test case. This figure shows that the largest difference in response due to a discontinuity does not determine the location of the peak IIF value. (**c**) $IIF^{R}$ values for three different window lengths of the STFT. The difference in time resolution and possible leaking in the STFT creates uncertainty in the exact peak location.

**Figure 5 entropy-22-01199-f005:**
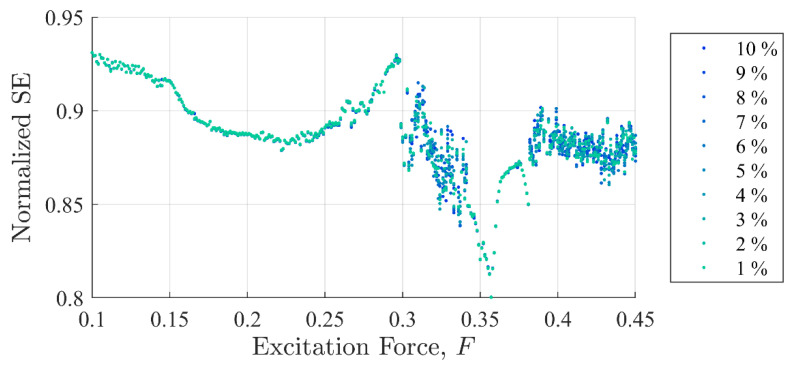
Shannon entropy values as a function of background excitation and discontinuity amplitude. The separation in the discontinuity amplitude test cases is not clear.

**Figure 6 entropy-22-01199-f006:**
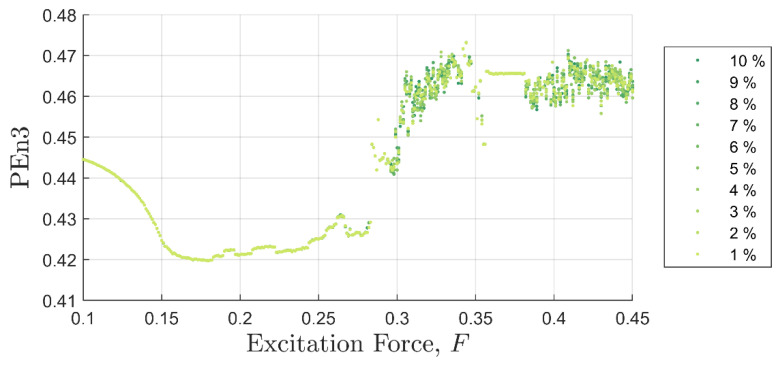
Permutation entropy values as a function of background excitation and discontinuity amplitude. There is no discernable separation in the discontinuity amplitude for most test cases.

**Figure 7 entropy-22-01199-f007:**
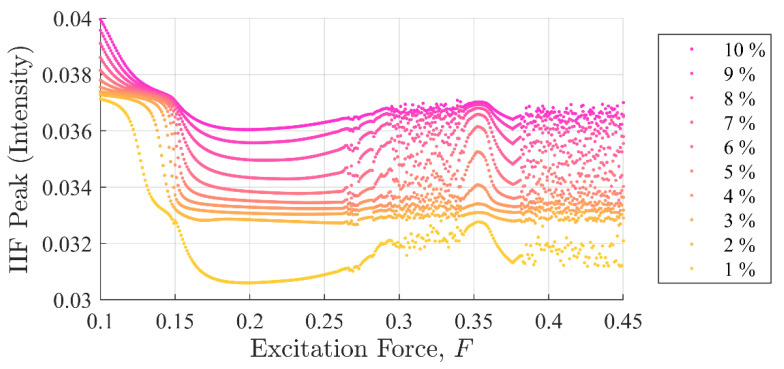
IIF peak values as a function of background excitation and discontinuity amplitude. Clear separation in the discontinuity amplitude test cases shows how well the IIF is able to detect a global disturbance even when the system is chaotic. The separation between amplitude test cases is in stark contrast to the SE and PEn results shown in Figure 5 and Figure 6.

**Figure 8 entropy-22-01199-f008:**
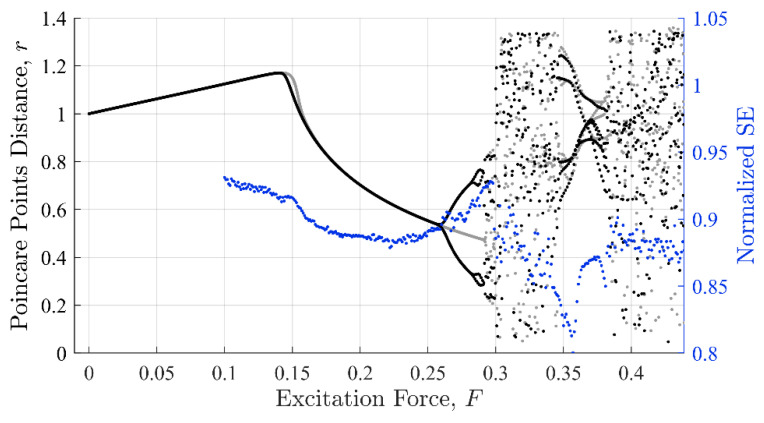
Shannon entropy against the Poincaré distance. The shape the SE values make as a function of excitation force mimics the Poincaré distance with similar inflection points near regions where the system behavior changes character.

**Figure 9 entropy-22-01199-f009:**
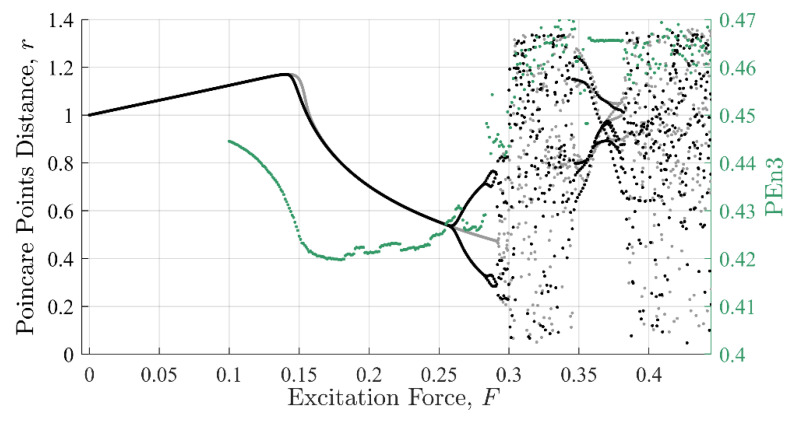
Permutation entropy against the Poincaré distance. The shape the PEn values make as a function of excitation force also mimics the Poincaré distance with similar inflection points. However, the shape loses form past F = 0.3.

**Figure 10 entropy-22-01199-f010:**
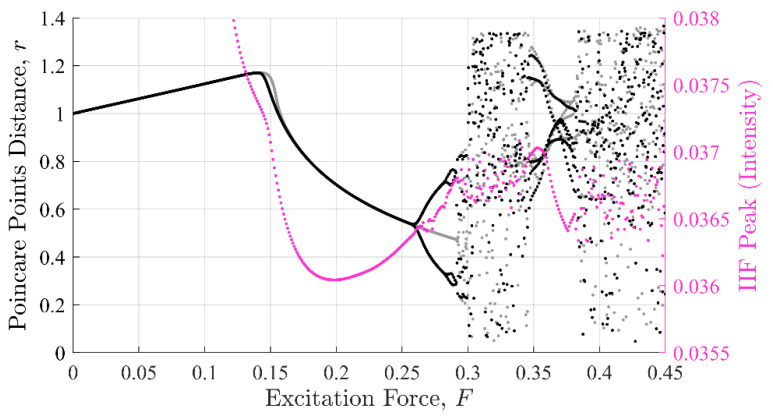
IIF peaks against the Poincaré distance. The shape the IIF peak values make as a function of excitation force mimics the Poincaré distance similarly to both the SE and PEn results, but with a smoother curve.

**Figure 11 entropy-22-01199-f011:**
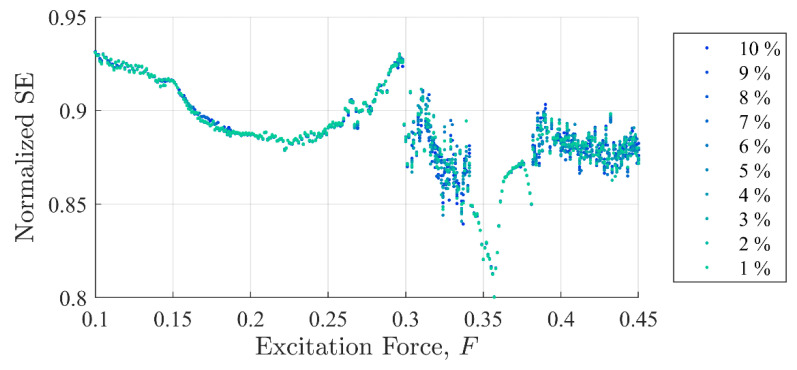
Shannon entropy values as a function of background excitation and discontinuity amplitude for case (ii). Inexplicably, values near F = 0.17 seem to show an increased entropy at 10% over lower values. Despite some small differences, these results closely follow those reported in Figure 6 for case (i).

**Figure 12 entropy-22-01199-f012:**
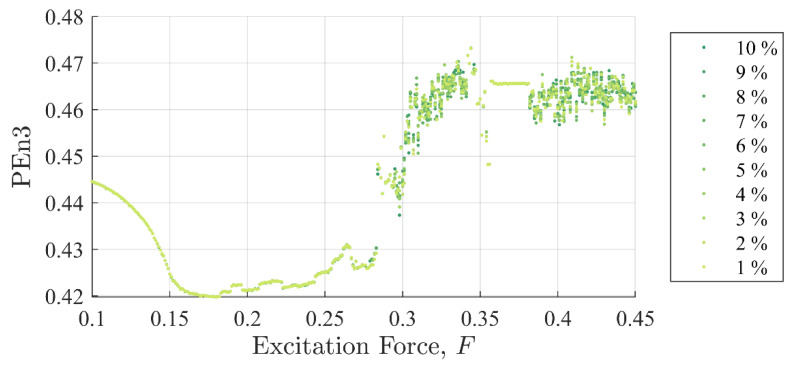
Permutation entropy values as a function of background excitation and discontinuity amplitude for case (ii). Like SE, these results closely follow those reported in Figure 7 for case (i).

**Figure 13 entropy-22-01199-f013:**
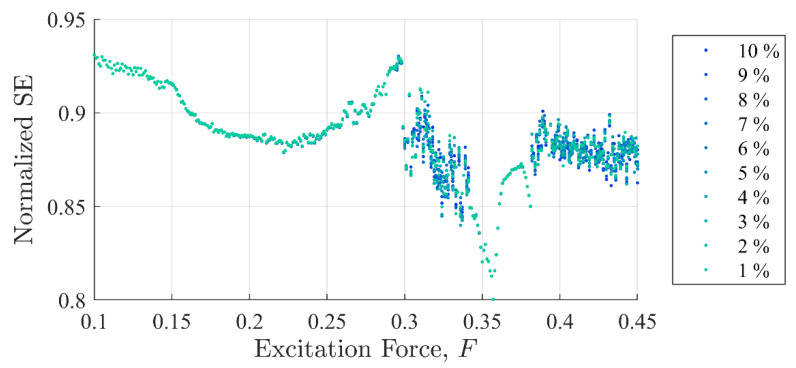
Shannon entropy values as a function of background excitation and discontinuity amplitude for case (iii). Again, these results closely follow those reported in Figure 5 for case (i) and Figure 11 for case (ii).

**Figure 14 entropy-22-01199-f014:**
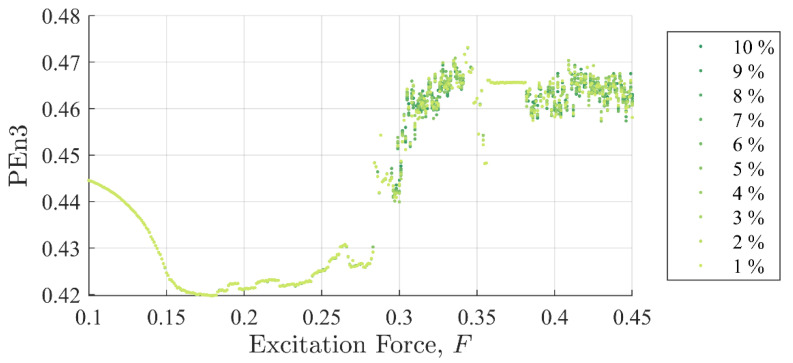
Permutation entropy values as a function of background excitation and discontinuity amplitude for case (iii). These results closely follow those reported for Figure 6 for case (i) and Figure 12 for case (ii).

**Figure 15 entropy-22-01199-f015:**
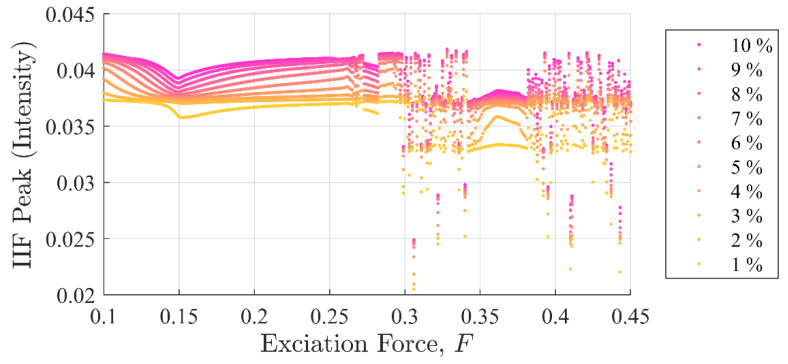
IIF peak values as a function of background excitation and discontinuity amplitude for case (ii). The separation in the discontinuity amplitude is apparent. The shape of the curves is very different from the global disturbance case due to the difference in sensitivity of the system to changes in stiffness. This was not the case for either SE or PEn.

**Figure 16 entropy-22-01199-f016:**
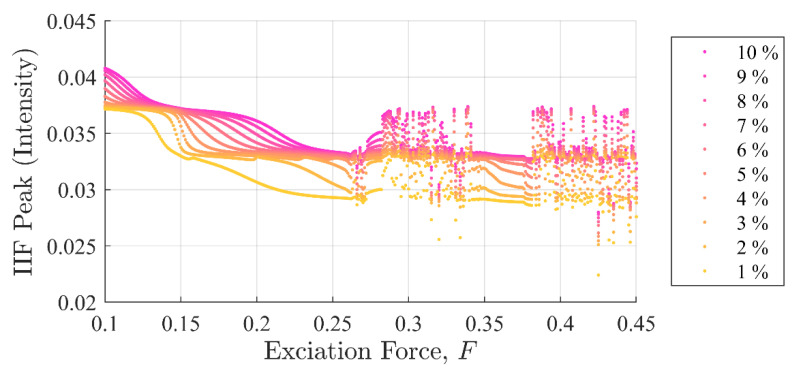
IIF peak values as a function of background excitation and discontinuity amplitude for case (iii). The separation in the discontinuity amplitude is apparent. The shape of the curves is very different from the global disturbance case and case (ii) due to the difference in sensitivity of the system to changes in damping. The difference in shape between case reported in Figure 7 for case (i), Figure 15 for case (ii) and the current image for case (iii) results from the sensitivity of the IIF to the response of the system.

**Figure 17 entropy-22-01199-f017:**
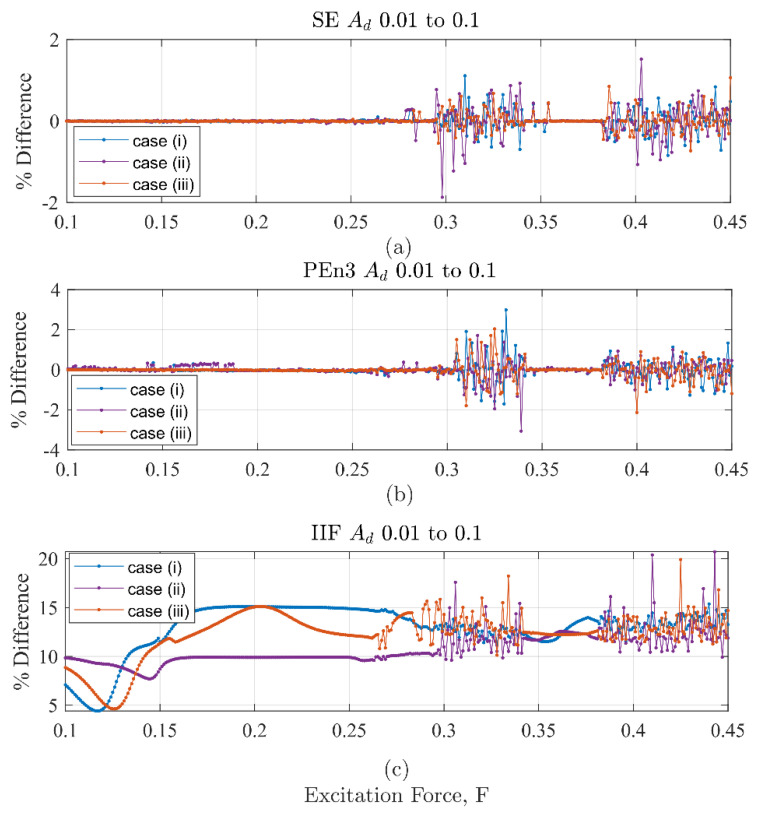
Difference values for cases (i), (ii), and (iii) as a function of excitation force. (**a**) SE results (**b**) PEn3 results (**c**) IIF results. The low values in (**a**) and (**b**) contrasted with the larger, consistently positive values in (**c**) show that the IIF is much better at distinguishing between the discontinuity levels.

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
