# Peer review of "Quantifying Information without Entropy: Identifying Intermittent Disturbances in Dynamical Systems"

_entropy, 2020, doi:10.3390/e22111199_

Round 1

Reviewer 1 Report

See attached text file.

Reviewer 2 Report

This paper describes a new method for quantifying the ramifications of localized perturbations of nonlinear systems using an information theoretic approach.  The connections between nonlinear dynamics of complex systems and the production of entropy has had a fruitful history in the field.  Here, the authors propose a novel formulation (the IIF method) that avoids the common framework of the Boltzmann entropy/Shannon information.  The theory is based on using time-frequency methodology combined with characterization by singular value decomposition.  Their approach is applied to the Duffing oscillator and the IIF analysis is compared to more conventional approaches.  The comparisons are favorable, establishing at least the promise of the new method as a tool for analyzing complex systems and data.

The paper is well-written and clear, and provides a nice but succinct review of the field as well as a detailed development of the formalism.  The work appears to be technically correct, and the subject and results are of interest to readers of the journal.  The work is significant and interesting, and their approach has the potential for high impact.  The article is acceptable for publication in its present form. 

Reviewer 3 Report

In their paper titled "Quantifying information without entropy: identifying intermittent disturbances in dynamical systems" authors study an important problem, and they put forward innovative solutions that can improve our ability to detect intermittent disturbances in dynamical systems -- an open problem with evergreen importance and appeal across many fields of research.

I have very much enjoyed reading this paper. I find it comprehensive and clearly written, and introducing new, timely, and important results concerning the detection of disturbances in dynamical systems. The following comments should be taken into account in a minor revision.

1) It would improve the paper if the figure captions would be made more self contained. In addition to what is shown for which parameter values, one could also consider a sentence or two saying what is the main message of each figure.

2) The quality of the figures looks to be too poor for production. The authors should supply high-quality figures that display well on screen and in print.

3) In the introduction, there is a recent research effort dedicated to entropy and prediction and the role of disturbances that would fit very well to the overview of relevant papers, namely Wavelet entropy-based evaluation of intrinsic predictability of time series, Chaos 30, 033117 (2020).

4) Also, it would be very useful if the authors would make their source code available as supplementary material. This would promote the usage of the proposed algorithm and allow also others to take advantage of this research.

5) Finally, let me note that a recent research effort showed explicitly that dynamical systems can be thought of as networks of interacting elements, and that this can give rise to chaos on its own right in Computational chaos in complex networks, J. Complex Netw. 8, cnz015 (2020) -- something that would also fit well in the first para. of the introduction.
